# Depressive Symptoms and Self-Esteem in White and Black Older Adults in the United States

**DOI:** 10.3390/brainsci8060105

**Published:** 2018-06-11

**Authors:** Shervin Assari, Maryam Moghani Lankarani

**Affiliations:** 1Department of Psychology, University of California Los Angeles (UCLA), Los Angeles, CA 90095, USA; 2Department of Psychiatry, University of Michigan, Ann Arbor, MI 48104, USA; lankaranii@yahoo.com; 3Center for Research on Ethnicity, Culture and Health, School of Public Health, University of Michigan, Ann Arbor, MI 48109-2700, USA

**Keywords:** race/ethnicity, ethnic groups, African Americans, evaluation of self, depressive symptoms

## Abstract

*Background.* Poor self-esteem is a core element of depression. According to recent research, some racial groups may vary in the magnitude of the link between depression and poor self-esteem. Using a national sample, we compared Black and White older Americans for the effect of baseline depressive symptoms on decline in self-esteem over time. *Methods.* This longitudinal study used data from the Religion, Aging, and Health Survey, 2001–2004. The study followed 1493 older adults (734 Black and 759 White) 65 years or older for three years. Baseline depressive symptoms (CES-D), measured in 2001, was the independent variable. Self-esteem, measured at the end of the follow up, was the dependent variable. Covariates included baseline demographic characteristics (age and gender), socioeconomic factors (education, income, and marital status), health (self-rated health), and baseline self-esteem. Race/ethnicity was the moderator. Linear multi-variable regression models were used for data analyses. *Results.* In the pooled sample, higher depressive symptoms at baseline were predictive of a larger decline in self-esteem over time, net of covariates. We found a significant interaction between race/ethnicity and baseline depressive symptoms on self-esteem decline, suggesting a weaker effect for Blacks compared to Whites. In race/ethnicity-specific models, high depressive symptoms at baseline was predictive of a decline in self-esteem for Whites but not Blacks. *Conclusion.* Depressive symptoms may be a more salient contributor to self-esteem decline for White than Black older adults. This finding has implications for psychotherapy and cognitive behavioral therapy of depression of racially diverse populations.

## 1. Introduction

Beck’s theory of depression defines depression as the negative evaluation of one’s self, others, and future [1]. Dysfunctional attitudes and beliefs about future, hopelessness, are strong determinants of suicidal behaviors [2,3]. This is supported by the extensive research that has documented lower levels of self-esteem, self-efficacy, mastery (evaluation of self) [4,5], relations quality (evaluation of others) [6], and hopelessness (evaluation of future) [7] among individuals with clinical or sub-clinical depression. These associations may, however, differ across various racial groups [8,9,10,11,12,13].

Although not all studies agree [8,9], a number of recent studies have suggested that Beck’s theory of depression is more relevant to Whites than Blacks [10,11,12,13]. In the presence of depression, Blacks may maintain higher levels of mastery (evaluation of self) and hope (evaluation of future) compared to Whites [12,13]. That is, depression has a smaller effect on positive cognitions and emotions in Blacks than Whites [13]. Racial differences also exist in the longitudinal associations between depressive symptoms and mastery [14,15,16,17,18,19,20,21,22]. Psychotherapy as well as modern pharmacological medications such as selective serotonin reuptake inhibitors (SSRIs e.g., fluoxetine) and serotonin-norepinephrine reuptake inhibitors (SNRIs e.g., duloxetine) demonstrate efficacy and potential to treat depression and prevent suicide, poor self-esteem, and hopelessness [23,24,25,26].

There are at least two studies on comparison of Blacks and Whites for the link between depression and self-esteem [8,9]. In the first study which included 1493 older individuals age 66 or more, depressive symptoms showed a larger association with poor self-esteem [8]. The second study used data of 3570 Blacks and 891 Non-Hispanic Whites who participated in the National Survey of American Life (NSAL), a nationally representative household survey, to investigate the reciprocal association between major depressive disorder (MDD) and low self-esteem. Similar to the first study, an interaction was found between race/ethnicity and MDD, suggesting a stronger association between depression and self-esteem among Blacks compared to Whites [9]. As both of these studies are limited due to cross-sectional design, there is still a need for additional research using longitudinal design.

Built on the Differential Effects hypothesis [27,28] and to replicate and extend the results of previous studies on this topic [8,9,10,11,12,13], this study explored racial variations in the association between depressive symptoms at baseline and subsequent decline in self-esteem among older Americans. Differential Effects hypothesis suggests that psychosocial risk and protective factors have smaller effects on behavioral and health outcomes for non-Whites (minorities) than Whites (the majority) [27,28]. We hypothesized that the effect of baseline depressive symptoms on decline in self-esteem over time would be smaller for Black than White older Americans.

## 2. Methods

### 2.1. Design and Setting

This longitudinal study used data from the Religion, Aging, and Health Survey (RAHS), 2001–2004, a prospective cohort study of older adults in the United States. The study followed participants for three years. The current study used data from Wave 1 and Wave 2 of the RAHS [29].

### 2.2. Ethics

The University of Michigan (UM) Institutional Review Board (IRB) approved the RAHS protocol. All participants provided informed consent.

### 2.3. Participants

Participants in the current study was limited to Black or White older Americans. Eligibility criteria were being non-institutionalized, English-speaking, age of 65 years or older at the time of enrollment. The study sample was limited to Christians or individuals who were not associated with any faith. The study oversampled Blacks, so almost half of the sample was Black. There were no additional exclusion criteria of this study. Due to the budget restrictions, the study sampling was restricted to the coterminous US (i.e., Alaska and Hawaii were not included) [29].

### 2.4. Sampling Frame

The study sampling was random. The sampling frame consisted of eligible individuals in the Medicare Beneficiary list. The list was obtained from the Centers for Medicare and Medicaid Services (CMS) at the time of survey in 2001. The study applied a five-step sampling procedure for drawing participants from the CMS file.

### 2.5. Data Collection

Data collection was performed by Louis Harris and Associates (Harris Interactive, Rochester, NY, USA). Wave 1 interviews were performed between March and August 2001 [29].

### 2.6. Study Measures

Race/ethnicity, demographic characteristics (age and gender), SES indicators (education, income, and marital status), overall health (self-rated health), depressive symptoms, and self-esteem were measured at Wave 1 in 2001.

*Sociodemographic Factors.* Demographic characteristics including age (continuous measure) and gender (1 female, 0 male) were measured. Socioeconomic status was measured using educational level (1 high school diploma, 0 lower education), marital status (1 married, 0 other), and person’s income (10 level categorical variable). For all SES indicators, a higher score was indicative of higher SES.

*Self-rated health (poor).* Individuals overall health status were measured using the following single item: “How would you rate your overall health at the present time?” Responses included: (1) Excellent, (2) Good, (3) Fair, and (4) Poor. We dichotomized the responses to excellent/good/fair 0 or poor 1. The single item SRH measure has shown high reliability and validity [30,31].

*Depressive Symptoms.* Depressive symptoms were measured using an 8-item Center for Epidemiological Studies-Depression scale (CES-D) [32]. The following negative emotions were evaluated: (1) the blues, (2) felt depressed, (3) crying spells, (4) feeling sad, (5) not feeling like eating (poor appetite), (6) feeling that everything is an effort, (7) restless sleep, and (8) could not get going. These items cover two domains of depression: negative affect and somatic symptoms. The 8-item CES-D has shown acceptable reliability and validity [33,34] and provides comparable results to the original CES-D with 20 items [35,36,37]. Response items had a range from (1) “rarely or none” to (4) “most or all of the time”. We calculated a mean depression score with a potential range from 1 to 4. This measure was treated as a continuous measure, with a higher score indicating more frequent symptoms of depression (Cronbach’s alpha = 0.87 for all, 0.85 for Whites, 0.89 for Blacks).

*Self-esteem*. Three items were borrowed from the Rosenberg Self-Esteem Scale, (RSES, Rosenberg, 1965) [38] to measure self-esteem (positive evaluation of self). The items included (1) “I feel I am a person of worth, or at least on an equal plane with others”, (2) “I feel I have a number of good qualities”, and (3) “I take a positive attitude toward myself”. The item responses were on a 4-point Likert scale: 1 (strongly disagree), 2 (disagree), 3 (agree), and 4 (strongly agree). All items were positively-worded. We calculated a sum score with a potential range from 3 to 12 with a higher score indicating more positive self-esteem. Several studies have used mean scores to define low and high self-esteem [39,40,41,42] (Cronbach’s alpha = 0.91 for all, 0.89 for Whites, 0.93 for Blacks).

### 2.7. Statistical Analysis

We used SPSS 22.0 for data analysis. To describe the sample overall, and also by race/ethnicity, we used mean (SD) and frequency tables. Pearson correlation and independent samples *t*-test were used for bivariate analysis. We used logistic regressions in the pooled sample, and specific to race/ethnicity, for our multivariable analysis.

We ran four linear regression models. The first two models were estimated in the pooled sample, and the last two models were specific to each race/ethnicity. *Model 1* did not include any interaction terms, however, *Model 2* did include an interaction term between race/ethnicity and depressive symptoms at baseline. In all models, baseline depressive symptoms were the independent variable (predictor). Self-esteem measured at baseline and end of follow up was the main dependent variable (outcome). Demographic factors, socio-economic status, health risk behaviors, and overall health at baseline were considered as the covariates. Race/ethnicity was the moderator. Adjusted regression coefficients, 95% confidence intervals (CIs), and *p* values were reported.

## 3. Results

The study followed 1493 older adults (age 65 or more), who were either Black (*n* = 734) or White (*n* = 759). From this number, 1011 individuals had pre and post data with a three-year interval.

### 3.1. Descriptive Statistics

Table 1 summarizes the descriptive statistics at baseline and three-year follow up data in the overall sample and by race/ethnicity. While Blacks and Whites were not significantly different by age, they differed in gender, with Black participants were more likely to be female than White participants. In addition, Blacks had lower SES as measured by education, income, and marital status than Whites. Compared to Whites, Blacks reported higher depressive symptoms at baseline. Blacks and Whites had comparable self-esteem at baseline and follow up.

### 3.2. Bivariate Correlations

Table 2 summarizes bivariate correlations in the pooled sample and by race/ethnicity. Depressive symptoms at baseline were associated with lower self-esteem in Whites, but not Blacks.

### 3.3. Models in the Pooled Sample

Table 3 summarizes the results of two linear regression models in the pooled sample. Based on *Model 1*, high depressive symptoms at baseline was predictive of a self-esteem decline in the pooled sample. *Model 2* showed a significant interaction between race/ethnicity and baseline depressive symptoms on decline in self-esteem, suggestive of a larger effect for Whites compared to Blacks.

### 3.4. Stratified Linear Regression Models by Race/Ethnicity

Table 4 summarizes the results of race/ethnicity-specific models on the association between baseline depressive symptoms and subsequent change in self-esteem among Black and White older adults. Based on *Model 3*, high depressive symptoms at baseline were predictive of a decline in self-esteem for White older adults. *Model 4*, however, did not show such an association for Black older adults.

## 4. Discussion

Using a national sample of older Americans, the current study documented considerable racial variation in the longitudinal association between baseline depressive symptoms and subsequent change in self-esteem over a three-year period. According to the results, depressive symptoms at baseline were predictive of a decline in self-esteem for White, but not Black, older adults.

The results of this study supported the Differential Effects hypothesis [27,28]. The results are relevant to Blacks’ self-esteem advantage, defined as higher self-esteem of Blacks compared to Whites [43]. Early work on Causal Attribution theory [44,45,46] has introduced system blame as a protective factor for self-esteem of Blacks. Blacks who blame the system instead of themselves for low social status are more likely to maintain positive emotions and cognitions about self (self-esteem) [47]. Blacks who attribute low social status to prejudice and racism can preserve self- esteem regardless of their relative social position [47]. This buffering effect of system blaming is absent for Whites, who may not easily attribute their low success to the system [48] (Please see [43] for a review). This research, however, is dated and may be less relevant at the current social/political climate.

Contemporary research has provided more insight on unique determinants of self-esteem in Blacks. This research has introduced racial and ethnic identity as a unique source of self-esteem for Blacks and other minority populations [49,50,51,52]. Other studies have shown that racial socialization and social support are strong determinants of self-esteem among Blacks [53]. Religion involvement and relationship with God also enhances self-esteem of Blacks [54]. This is particularly important as protective effects of social relations and religiosity are stronger for Whites than Blacks [55,56,57,58,59,60,61,62]. These studies collectively suggest that developmental processes that shape self-esteem, may widely differ for White and Black individuals [63]. That is social and psychological processes that impact self-esteem development are specific to sub-populations [63]. Contribution of race/ethnicity specific determinants of self-esteem of Blacks may potentially explain why in this study baseline depressive symptoms did not predict a decline in self-esteem for Blacks. The salience of these processes in shaping Blacks’ self-esteem, however, is probably differently relevant across different cohorts who experience different political climates.

Advantage of Blacks in having larger health gains from social support and religion compared to Whites [55,56,57,58,59,60,61,62] is related to Blacks disadvantage in gaining health from economic resources, which is probably a healthy adjustment to the adversities and scarcity of economic resources in their lives [14,27,28,64,65,66,67,68,69,70,71,72,73,74]. So, we argue that these differences may be result of healthy adjustments in Blacks compared to Whites.

These results are also in line with the previous studies have suggested that Blacks and Whites differ in the domains that impact self-esteem [75]. For instance, Whites may have a higher tendency to base their self-esteem on the approval of others than Blacks [75].

Our results in Black older adults are similar to the results of a study among 208 Black single mothers, aged 18 to 45, who were recruited from community settings. In that study, self-esteem was not linked to depressive symptoms, when anger, perceived stress and perceived racism were controlled [76]. Several other studies, however, have shown the effects of depression [77,78,79,80] and distress [78] on self-esteem in Blacks.

In several studies, depressive symptoms [81] and other indicators of negative affect, such as neuroticism [22], are shown to be associated with an increased subsequent risk of MDD for Whites, but not Blacks. Black-White differences in the predictive role of negative affect on future risk of MDD may explain why one-time measurement of negative affect does not reflect the same level of depression risk for Black and White individuals. Depressive symptoms and negative affect better reflect future risk of negative affect for Whites than Blacks [22,81].

Although Beck’s cognitive model of depression suggests that depression accompanies negative evaluations of self, others, and the future [1], the relevance of Beck’s theory to depression may be different for Blacks and Whites. Although there is considerable research showing an effect of depression on self-esteem [4] and mastery [5], relation quality with others [6], and hopelessness [7], these effects are not universal across races/ethnicities. For instance, depression differently impacts positive emotions and cognitions such as self-esteem [4,8,9], hope [12], positive emotions [13], and mastery [10,11] in diverse racial groups. For instance, depression has smaller effects on mastery [10,11] and hope [12] in Blacks than Whites.

Findings of the current study may have implications for understanding racial differences in the health effects of depression and negative emotions in racially diverse populations of older adults. In several studies, race/ethnicity is shown to alter the effects of depressive symptoms and health domains such as obesity [82,83,84], cardiovascular diseases (CVDs) [85], chronic disease [15], and mortality due to all-causes, cardiovascular disease, or renal disease [16,86]. Depression is also differently linked to inflammation between Whites and Blacks [87,88]. One explanation for smaller effects of depression on medical conditions is the undoing hypothesis, which is the reduction of the negative medical consequences of depression in the presence of positive emotions and cognitions [89,90].

These findings refute a “one size fits all” outlook, as Blacks and Whites vastly differ in the associations between a wide range of processes that shape their psychosocial well-being. The association between depression and socioeconomic status [64,65,91], stress [92], behaviors [93,94], and health [15,82,85,95,96], differ for Whites and Blacks. These findings suggest that differential effects are rules rather than exceptions between Blacks and Whites [27,28].

### Limitations

Our study had a few limitations. The first limitation is potential measurement bias. We used only few items to measure self-esteem and depressive symptoms, which may not have the ideal validity and reliability. More standard measures with a greater number of items are needed to replicate these findings. Measures of depressive symptoms and self-esteem may have different validity by race/ethnicity. We, however, do not attribute these findings to measurement bias, as depressive symptoms reflect concurrent risk of depression in Blacks and Whites [81,97]

The second limitation is due to omitted variables. The current study failed to control for some of the confounders such as physical health, social support, religion, stress, and health care access. The third limitation is that this study only included Blacks and Whites. There is a need for future studies on other racial and ethnic groups. Finally, the only socioeconomic covariates were education and marital status. Other indicator such as occupation, employment, wealth, income, and household size were not included. A strength of the current study was a national sample. As the results were specific to adults aged 65 and older, the results should not be generalized to other age groups of Blacks and Whites.

## 5. Conclusions

In the current study, Black and White older adults showed that they may differ in the magnitude of the effects of baseline depressive symptoms on subsequent change in self-esteem over time. Clinicians should be aware that depressive symptoms impact self-esteem of Whites but not Blacks. This finding is in line with previous findings on differential relevance of Beck’s theory for depression for Blacks and Whites. White older adults’ depression predicting future decline in positive evaluation of self may suggest that psychotherapy of depression may have an impact for evaluation of self for Whites but not Blacks.

## Figures and Tables

**Table 1 brainsci-08-00105-t001:** Descriptive statistics in the pooled sample and by race/ethnicity.

	All (*n* = 1439)		Whites (*n* = 759)		Blacks (*n* = 734)	
	Mean	SD	Mean	SD	Mean	SD
Age	75.14	6.67	75.37	6.82	74.91	6.49
Income *^a^	4.59	2.49	5.63	2.49	3.49	1.96
Depressive Symptoms *^a^	1.56	0.62	1.54	0.59	1.59	.65
Self-Esteem 1	10.36	1.46	10.13	1.41	10.61	1.47
Self-Esteem 2	10.50	1.54	10.45	1.53	10.55	1.56
	*N*	*%*	*n*	*%*	*N*	*%*
Gender *^b^						
Male	573	38.20	314	41.37	256	34.88
Female	927	61.80	445	58.63	478	65.12
Education *^b^						
Low	609	40.98	200	26.60	407	55.98
High	877	59.02	552	73.40	320	44.02
Married *^b^						
No	778	52.28	306	40.53	467	64.33
Yes	710	47.72	449	59.47	259	35.67
Self-Rated Health (SRH) Poor *^b^						
No	1322	88.37	694	91.80	622	84.86
Yes	174	11.63	62	8.20	111	15.14

**Source:** Religion, Aging, and Health Survey, 2001–2004. * *p* <0.05. ^a^ independent samples *t* test. ^b^ Chi square test.

**Table 2 brainsci-08-00105-t002:** Bivariate correlations between baseline and subsequent variables in the pooled sample and by race/ethnicity.

	1	2	3	4	5	6	7	8	9
All (*n* = 1439)									
1 Race/ethnicity (Black)	1	0.07 **	−0.03	−0.21 **	−0.24 **	−0.02	0.05	0.17 **	0.03
2 Gender (Female)		1	0.02	−0.11 **	−0.32 **	−0.02	0.08 **	0.02	0.03
3 Age			1	0.00	−0.18 **	0.08 **	0.03	−0.06 *	−0.06
4 Education (College Graduation)				1	0.13 **	−0.05	−0.12 **	0.07 **	0.08 **
5 Marital Status					1	−0.01	−0.12 **	−0.03	0.04
6 Self-Rated Health (SRH)						1	0.24 **	−0.55 *	−0.06
7 Depressive Symptoms							1	−0.17 **	−0.10 **
8 Self-Esteem 1								1	0.17 **
9 Self-Esteem 2									1
Whites									
2 Gender (Female)		1	0.08 *	−0.14 **	−0.32 **	0.00	0.10 **	0.03	−0.02
3 Age			1	0.02	−0.24 **	0.10 ^**^	0.08 *	−0.05	−0.09 *
4 Education (College Graduation)				1	0.07	−0.05	−0.15 **	0.14 **	0.11 *
5 Marital Status					1	−0.07	−0.15 **	−0.03	0.06
6 Self-Rated Health (SRH)						1	0.24 **	−0.05	−0.07
7 Depressive Symptoms							1	−0.13 **	−0.17 **
8 Self-Esteem 1								1	0.19 **
9 Self-Esteem 2									1
Blacks									
2 Gender (Female)		1	−0.03	−0.03	−0.31 **	−0.04	0.06	−0.01	0.08
3 Age			1	−0.05	−0.14 **	0.07	−0.01	−0.05	−0.02
4 Education (College Graduation)				1	0.12 **	−0.05	−0.05	0.08 *	0.08
5 Marital Status					1	0.04	−0.08 *	0.04	0.03
6 Self-Rated Health (SRH)						1	0.23 **	−0.05	−0.05
7 Depressive Symptoms							1	−0.23^**^	−0.05
8 Self-Esteem 1								1	0.14 **
9 Self-Esteem 2									1

**Source:** Religion, Aging, and Health Survey, 2001–2004. * *p* <0.05. ** *p* <0.001.

**Table 3 brainsci-08-00105-t003:** Association between baseline depressive symptoms (2001) and subsequent change in self-esteem (2001–2004) using linear regression models in the pooled sample.

	All (*n* = 1439)	95% CI	*p*	All (*n* = 1439)	95% CI	*p*
	*Model 1*			*Model 2*		
	OR			OR		
Race/ethnicity (Black)	0.14	−0.07–0.34	0.199	−0.43	−0.98–0.12	0.127
Age	−0.02	−0.03–0.00	0.065	−0.01	−0.03–0.00	0.073
Gender (Female)	0.15	−0.06–0.36	0.173	0.14	−0.07–0.36	0.181
Education	0.34	0.06–0.61	0.016	0.32	0.05–0.59	0.021
Marital Status (Married)	0.12	−0.10–0.33	0.286	0.11	−0.11–0.32	0.327
Self-Rated Health (SRH)	−0.22	−0.68–0.23	0.339	−0.23	−0.69–0.22	0.314
Self-Esteem 1	0.16	0.09–0.23	<.001	0.17	0.10–0.24	<0.001
Depressive Symptoms	−0.19	−0.37–0.02	0.029	−0.39	−0.63–0.14	0.002
Depressive Symptoms × Race/ethnicity	-	-	-	0.37	0.04–0.70	0.030
Intercept	9.99	8.48–11.49	<0.001	10.21	8.70–11.73	<0.001

**Source:** Religion, Aging, and Health Survey, 2001–2004. Outcome: Self-Esteem 2.

**Table 4 brainsci-08-00105-t004:** Association between baseline depressive symptoms (2001) and subsequent change in self-esteem (2001–2004) using linear regression models among Whites and Blacks.

	Whites (*n* = 759)			Blacks (*n* = 734)		
	*Model 3*	95% CI	*p*	*Model 4*	95% CI	*p*
	OR			OR		
Age	−0.03	−0.05–0.00	0.020	0.00	−0.02–0.02	0.940
Gender (Female)	0.05	−0.24–0.33	0.750	0.29	−0.03–0.62	0.079
Education	0.31	−0.01–0.63	0.057	0.37	−0.16–0.91	0.171
Marital Status (Married)	0.05	−0.24–0.35	0.724	−	−0.13–0.51	0.246
Self-Rated Health (SRH)	−0.10	−0.70–0.51	0.756	−0.38	−1.08–0.32	0.285
Self-Esteem 1	0.17	0.08–0.27	<0.001	0.16	0.05–0.26	0.003
Depressive Symptoms	−0.39	−0.64–0.14	0.002	−0.01	−0.26–0.23	0.915
Intercept	11.10	9.03–13.16	<0.001	8.73	6.51–10.94	<0.001

**Source:** Religion, Aging, and Health Survey, 2001–2004. Outcome: Self-Esteem 2.

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
