# Peer review of "Depressive Symptoms and Self-Esteem in White and Black Older Adults in the United States"

_brainsci, 2018, doi:10.3390/brainsci8060105_

Round 1

Reviewer 1 Report

This is, in summary, an interesting paper aimed to compare Black (N=734) and White (N=759) older Americans for the effect of baseline depressive symptoms on decline in self-esteem over time. The authors found that in the pooled sample, higher depressive symptoms at baseline were predictive of a larger decline in self-esteem over time, net of covariates. They also reported a significant interaction between race and baseline depressive symptoms on self-esteem decline, suggesting a weaker effect for Blacks compared to Whites. In addition, in race-specific models, high depressive symptoms at baseline resulted a significant predictor of a decline in self-esteem for Whites but not for Blacks.

The authors may find as follows my main comments/suggestions.

First, as the authors, within the Introduction section, referred to the importance of hopelessness in terms of depression, they could also mention the relevance of this concept in terms of suicide risk given that the presence of hopelessness has been widely recognized as an important and independent risk factor for suicide (which is frequently linked to depression).

In addition, as the authors correctly mentioned, throughout the same section, the relevance of depressive symptoms and poor self-esteem in their paper, they could also report that modern pharmacological medications such as SSRIs and more recent SNRIs (e.g., duloxetine) demonstrated to date their efficacy and their potential to prevent negative consequences associated with major depression. The correct and rapid recognition/treatment of this disabling condition is an absolute imperative for the whole community. The improvement of major depression patient medical care is hardly necessary together with the identification of predictors that could help in diagnosis, classification of subtypes and monitoring of disease progression. Thus, in order to briefly address this issue (although i understand that this is not the main topic of the present manuscript), i suggest to cite and discuss the study published on Hum Psychopharmacol in 2009 (PMID: 19229839).

Furthermore, considering that the main aims of this paper have been extensively proposed by the authors, the specific hypotheses underlying the study objectives could be adequately reported as well.

Moreover, whether the local review board regularly approved the main study design needs to be specified.

Also, the authors sould more carefully define and report the most relevant inclusion/exclusion criteria of this study.

In addition, throughout the Discussion section some statements such as “developmental processes that shape self-esteem, may widely differ for Whites and Blacks” or “depression differently impacts self-esteem, hope, positive emotions, and mastery in diverse racial groups” are interesting as presented but need to be better developed.

Importantly, the fact that self-esteem, depressive symptoms, and health have been investigated only using some items of specific self-rated questionnaires and not specific ad hoc psychometric measures by participants need to be inserted within the main limitations/shortcomings of the manuscript. 

Finally, what is the take-home message of the present manuscript? While the authors clearly specified that findings of the current study may have implications for understanding racial differences in the health effects of depression and negative emotions in racially diverse populations of older adults, they failed, in my opinion, to depict the most relevant conclusive remarks for clinicians according to their findings.

Author Response

The changes include :

1- some changes to the introduction on relevance of hopelessness for suicide

2- A sentence on medications added to the introduction.

3- adding the hypothesis to the end of the introduction.

4- explained the inclusion / exclusion criteria

5- The IRB section is added.

6- I added a bit on social processes that impact self-esteem.

7- Added to the limitation section.

8- Revised the conclusion (take home message)

Reviewer 2 Report

Interesting research article on the differences in depressive symptoms between Black and White older adults.

As minor issue, authors could add more in limitations section.

In example, did lower education and income affect the results?

Author Response

Added some text to the limitations on SES that could be better examined.

Round 2

Reviewer 1 Report

In the revised manuscript, the authors generally addressed most of the major questions raised by Reviewers improving the main structure of the paper. I have no further additional comments/suggestions